# Three-dimensional numerical simulation of mud flow from a tailings dam failure across complex terrain

Dayu Yu[1,2], Liyu Tang[1,2], Chongcheng Chen[1,2]

[1]Key Laboratory of Spatial Data Mining & Information Sharing of Ministry Education, Fuzhou University, Fuzhou 350108, China

[2]National Eng. Research Center of Geospatial Information Technology, Fuzhou University, Fuzhou 350108, China

*Correspondence to*: Liyu Tang (tangly@fzu.edu.cn)

**Abstract.** A tailings dam accident can cause serious ecological disaster and property loss. Simulation of a tailings dam accident in advance is useful for understanding the tailings flow characteristics and assessing the possible extension of the impact area. In this paper, a three-dimensional (3-D) computational fluid dynamics (CFD) approach was proposed for reasonably and quickly predicting the flow routing and impact area of mud flow from a dam failure across 3-D terrain. The Navier–Stokes equations and the Bingham-Papanastasiou rheology model were employed as the governing equations and the constitutive model, respectively, and solved numerically in the finite volume method (FVM) scheme. The volume of fluid (VOF) method was used to track the interface between the tailings and air. The accuracy of the CFD model and the chosen numerical algorithm were validated using an analytical solution of the channel flow problem and a laboratory experiment on the dam break problem reported in the literature. In each issue, the obtained results were very close to the analytical solutions or experimental values. The proposed approach was then applied to simulate two scenarios of tailings dam failures, one of which was the Feijão tailings dam that failed on January 25, 2019, and the simulated routing coincided well with the in situ investigation. Therefore, the proposed approach does well in simulating the flow phenomenon of tailings after a dam break, and the numerical results can be used for early warning of disasters and emergency response.

## 1. Introduction

A tailings dam is one of the three basic projects at a mine, and it is a dangerous source of artificial mud flow with high potential energy. The stability of tailings dams is highly susceptible to natural and human-induced adverse factors. According to the International-Commission-on-Large-Dams (ICOLD) (Dams, 2001), the accident rate of tailings dams in the past 100 years (1900-2000) was approximately 1.2%, which is two orders of magnitude higher than the accident rate of reservoirs (0.01%). Since the beginning of the 20th century, there have been hundreds of tailings dam accidents in the world (Rico et al., 2008b). In 2008, the Xiang Fen tailings dam collapsed in China, causing 381 deaths. In 2015, the Fundão mine tailings dam failure in Brazil released 32 million $m^3$ of toxic tailings, polluted 650 km of rivers and flowed into the Atlantic Ocean, causing serious environmental consequences (Burritt and Christ, 2018). Four years later, more than 232 people died in the

failure of the Feijão tailings dam, which belongs to the same company as the Fundão mine (Santamarina et al., 2019).

To date, the number of tailings dams in the world exceeds $10^5$, and ensuring the stability of such reservoirs is an extremely challenging task for the mining industry. By documenting 147 historical cases of tailings dam failure worldwide, Rico et al. (2008b) found that the most common causes of failure are related to unusual rain and seismic liquefaction. Furthermore, dozens of factors, such as management operations, slope instability, fluvial undermining, and foundation failure, are also likely to cause dam failures. As a
consequence, even if all the tailings dams were constructed correctly, the occurrence of pond failure cannot be avoided.

Dam-break simulation and run-out analyses can provide useful information for assessing the risk associated with tailings dam collapses, such as run-out path and quantification of potential losses, which have become
more important in the geotechnical assessment of tailings dams. Nevertheless, little work has been conducted to analyze the dam-break process and investigate the flow of tailings released following dam failure. These related dam-break analyses can be divided into empirical models and numerical models. The empirical model predicts the volume of released tailings ($V_F$) and the run-out distance ($D_{max}$) by establishing a regression equation based on historical cases of collapse, dam height, and the impounded volume of tailings (Rico et al.,
2008a;Larrauri and Lall, 2018). However, owing to the difference in the actual state of the tailings reservoir and the surrounding environment, and without considering the rheological properties of the tailings such as the viscosity coefficient and yield stress (Martin et al., 2015), the predicted results may deviate greatly from the actual dam-break situation. Additionally, the flow pattern and movement regularity of tailings fluid cannot be accurately described.

Since numerical modeling based on computational fluid dynamics (CFD) has the advantages of sophisticated theory and low cost, it is considered to be a more appropriate solution for analyzing geographic hazards such as water reservoir failures, landslides and debris flows (Dai et al., 2014;Issakhov and Imanberdiyeva, 2019;Han et al., 2019). Using CFD to study and analyze a tailings run-out path is rare. In many cases, the
depth-averaged shallow water model has been widely used to compute the fluxes on rectangular grids. It can be numerically solved by different schemes such as smoothed particle hydrodynamics (SPH) (Cascini et al., 2014), finite volume method (FVM) (Medina et al., 2008;Li et al., 2013;Wang et al., 2020), finite difference method (FDM) (Wu et al., 2013). However, the shallow water equations are two-dimensional simplifications of Navier-Stokes equations, neglecting the observed nonlinear behavior through the mud depth (Han et al.,
2019). For more reliable results, the 3-D Navier-Stokes equations are recommended to control the flow dynamic (Wang et al., 2016;Lee et al., 2010).

One key issue limiting the application of CFD method in tailings flow simulation is the rheological properties. Unlike water, which is a Newtonian fluid, the tailings fluid formed by mixing tailings and water is in nature a
non-Newtonian fluid (Henriquez and Simms, 2009) with complex rheological properties. In the early research, the Voellmy-frictional model, frictional model, and the general two-phase model were used to describe the rheological behavior of a mixture of water and clay-like tailings fluids and were successfully applied in 2-D

shallow water equations, but these constitutive models may present a difficulty in discretizing in the 3-D FVM scheme(Han et al., 2019). Currently, the Bingham model is considered a better way to describe the dynamic behavior of tailings fluids and easy to apply in 3-D case (Mizani et al., 2013;Henriquez and Simms, 2009;Liao and Zhou, 2015;Gao and Fourie, 2015). Another key factor that makes it difficult to analyze and predict tailings run-out routing by numerical modeling is that the difficulty of building the boundary of large-scale complex downstream terrains. Recently, some initial work, such as laboratory experiments and relatively simple deposition simulations of tailings fluid, has been attempted (Mizani et al., 2013;Zhang et al., 2015;Babaoglu and Simms, 2017;Gao and Fourie, 2019;Luo et al., 2017). These studies discussed the flow and rheological properties of tailings fluids in detail but did not take into account the influence of real terrain on the movement distance of tailings fluids. Wang (Wang et al., 2018) integrated the SPH and the 30-meter resolution digital elevation model (DEM) of the ALOS satellite to simulate the dam-break process of a tailings reservoir, while the complex rheological properties of the tailings fluids were not mentioned at all.

In this paper, a three-dimensional (3-D) model based on FVM method for simulating mud flow from a tailings dam break across 3-D terrain was presented to provide quantitative information for risk assessment and disaster prevention, and the accuracy of the model was tested using small-scale laboratory experiments of a dam-break type flow. Aiming at the difficulty of accurately reproducing large-scale downstream terrain, unmanned aerial vehicle (UAV) aerial photogrammetry technology was used to obtain fine DEM data. Furthermore, the calculation of the model was efficiently executed on the supercomputer cluster of the Fujian high-performance computing (HPC) center. Finally, we simulated two scenarios of tailings dam failure.

## 2. Methodology

### 2.1. Mathematical and numerical models

In this study, the mixture of tailings and water is regarded as an incompressible fluid, and the tailings-flow dynamic behavior can be described by the 3-D continuity and Navier-Stokes (N-S) governing equations, which follow the physics laws of conservation of mass and momentum. The governing equations (Versteeg and Malalasekera, 2007) can be written in the tensor form as:

$$\frac{\partial \rho}{\partial t} + \frac{\partial}{\partial x_i}(\rho u_i) = 0 \tag{1}$$

$$\frac{\partial}{\partial t}(\rho u_i) + \frac{\partial}{\partial x_j}(\rho u_i u_j) = -\frac{\partial p}{\partial x_i} + \frac{\partial}{\partial x_j}\left(\mu_e\left(\frac{\partial \overline{u_i}}{\partial x_j} + \frac{\partial \overline{u_j}}{\partial x_i}\right)\right) \tag{2}$$

where $\rho$ is the density; $t$ is the time; $u_i$ is the velocity; $p$ is the pressure; $\mu_e$ is the effective viscosity.

Most natural or engineering fluids are in a turbulent state that causes fluctuations in velocity, pressure, temperature and other transport scalars in space and time. Tailings reservoirs are mostly located in mountainous areas, with a large terrain gradient. Under the influence of gravity, tailings fluids flow out of the break at high speed and will be in a turbulent state (Blight, 1997). The standard k-ε Reynolds-Averaged Navier–Stokes (RANS) equations (Jones and Launder, 1972) are used to address the turbulence effect. This method introduces the kinetic energy k and the turbulent dissipation rate ε with good convergence, which is the most widely used method in the engineering field. The RANS equations can be written as:

$$\frac{\partial}{\partial t}(\rho \overline{u_i}) + \frac{\partial}{\partial x_j}(\rho \overline{u_i u_j}) = -\frac{\partial \overline{p}}{\partial x_i} + \frac{\partial}{\partial x_j}\left(\mu_e\left(\frac{\partial \overline{u_i}}{\partial x_j} + \frac{\partial \overline{u_j}}{\partial x_i}\right) - \rho \overline{u'_i u'_j}\right) \tag{3}$$

where — denotes the average value; and $'$ denotes the pulsation value.

In the present model, the tailings fluid and air are assumed to be two incompressible phases, and the surface of the tailings phase movement is captured using the volume-of-fluid (VOF) method (Hirt and Nichols, 1981) that uses a step function $F$ whose value is unity at any control volume occupied by tailings fluid and zero otherwise. The time dependence of $F$ is governed by the eq. (4):

$$\frac{\partial F}{\partial t} + \nabla \cdot (u_i F) = 0 \tag{4}$$

The above governing equations are solved based on the FVM, which is the most commonly used method in the field of fluid engineering (Schraml et al., 2015;Yoon et al., 2014). We split the computational region into series of control volumes, which is represented by a mesh of hexahedral cells in the Eulerian framework according to the FVM. The governing equations are discretized into a set of algebraic expressions involving the unknown values of the physical quantities at the center of each control volume by integrating the N-S equation over the control volume. The following high-order discretization schemes (Table 1) are used to interpolate the physical quantities of the centers of the control volumes onto the faces, and this setup is second-order accurate and fully bounded.

**Table 1.** Discretization schemes of the model

| $t$ | $\nabla$ | $\nabla \cdot$ | $\nabla^2$ |
|---|---|---|---|
| Euler | Gauss linear | Gauss vanLeer/ Gauss linear | Gauss linear corrected |

where $\nabla$ is the Gradient operator; $\nabla \cdot$ is the Nabla operator; $\nabla^2$ is the Lapacian operator.

Preconditioning conjugate gradients (PCG) and smooth linear algebraic solvers are selected to calculate symmetric and asymmetric matrices separately in the work. Additionally, in terms of pressure-velocity coupling, we use the pressure implicit with the splitting of operators (PISO) (Issa et al., 1991) algorithm to achieve it. The PISO algorithm uses prediction-correction-recorrection steps, speeding up the convergence speed in the iterative process and improving computational efficiency. To guarantee model stability, the Courant-Friedrichs-Lewy (CFL) condition must be satisfied. For the $n$-dimensional case, the CFL number is defined in the following general form:

$$CFL = \Delta t \sum_{i=1}^{n} \frac{u_i}{\Delta x_i} \tag{5}$$

The CFL number is a measure of how much information ($u_i$) traverses a control volume ($\Delta x_i$) in a given time-step ($\Delta t$). The CFL condition can be restricted by a maximum CFL coefficient between 0 and 1, which is chosen to be 0.5 in the present work.

### 2.2. Rheological behavior of tailings fluid

Tailings fluid is similar to debris flow and is generally considered an incompressible and viscous fluid. Currently, most existing studies on debris flows and landslides assume that the mud-water mixture is a

homogeneous non-Newtonian fluid with a constant density (Dai et al., 2014;Wang et al., 2016;Han et al., 2019). Several authors concluded that a fluid with solid concentrations above 10% is a non-Newtonian fluid (Komatina and Jovanovic, 1997;Sloff, 1993), and their rheological properties can be described by the Bingham model (Pastor et al., 2014;Komatina and Jovanovic, 1997;Han et al., 2019) that can be represented by the following constitutive equation:

$$\tau_{ij} = (\frac{\tau_b}{\dot{\gamma}} + \mu_b)\dot{\gamma}_{ij} \tag{6}$$

where $\tau_b$ is the yield stress; $\mu_b$ is the viscosity coefficient; $\tau_{ij}$ is the shear stress tensor; $\dot{\gamma}_{ij}$ is the rate-of-deformation tensor:

$$\dot{\gamma}_{ij} = \frac{\partial u_i}{\partial x_j} + \frac{\partial u_j}{\partial x_i} \tag{7}$$

The symbol $\dot{\gamma}$ is the second invariant of the $\dot{\gamma}_{ij}$, which is given by:

$$\dot{\gamma} = \sqrt{\frac{1}{2}\dot{\gamma}_{ij}\dot{\gamma}_{ij}} \tag{8}$$

Bingham fluid is a kind of plastic fluid; as long as the shear stress exceeds the threshold $\tau_b$ specific to the material, the material flows like a Newtonian fluid. Where the threshold is not exceeded, the plastic fluid behaves like a solid. The average diameter of the tailings is generally less than 0.1 mm. At present, several laboratory rheological tests have shown that the rheological properties of a water-tailings mixture conform to those of a Bingham fluid (Mizani et al., 2013;Henriquez and Simms, 2009;Liao and Zhou, 2015;Gao and Fourie, 2015); hence, the Bingham model is adopted as the constitutive model to describe the fluidization characteristics of flow-like tailings in this work. In the Bingham model, the unknown parameters are $\tau_b$ and $\mu_b$, which can be determined by laboratory measurements. The apparent viscosity in a Bingham model can be represented as:

$$\mu_e = \frac{\tau_b}{\dot{\gamma}} + \mu_b \tag{9}$$

Understandably, the effective viscosity will become unbound when $\dot{\gamma}$ tends to be infinitely small, such as in the core region of a Bingham flow, which may cause numerical divergence or even crash in the solution procedure (Shao and Lo, 2003). To overcome this inherent discontinuity issue, several regularized approaches have been proposed, where the infinite viscosity presented in the rigid zone can be approximated by a highly viscous fluid (Papanastasiou, 1987;Hannani et al., 2007;Frigaard and Nouar, 2005). The most popular regularization is the Bingham-Papanastasiou model(Papanastasiou, 1987), which is chosen as the constitutive equation of tailings fluid in the present work. Hence, the apparent viscosity is expressed as:

$$\mu_e = \frac{\tau_b}{\dot{\gamma}}[1 - e^{(-q\dot{\gamma})}] + \mu_b \tag{10}$$

where $q$ denotes the stress growth parameter, such that exceeding the yield stress $\tau_b$ finite shear stress is allowed for small shear rates, while the stress grows linearly with shear stress beyond the yield stress. Moreover, the limit of $m \to \infty$ is fully equivalent to an ideal Bingham fluid.

In this study, mathematical and rheological models are numerically built and implemented using the release of version 6 (https://github.com/OpenFOAM/OpenFOAM-6) of the open-source Field Operation and
Manipulation (OpenFOAM) C++ libraries under a Linux environment. OpenFOAM is a framework for developing application executables that use packaged functionality contained within a collection of approximately 100 C++ libraries (Greenshields, 2018). In the original OpenFOAM code, it is impossible to cope with the Bingham flow. To make it possible for the tailings-flow simulation, the regularized Bingham-Papanastasiou model without numerical divergence was introduced into the original code.

**2.3. Experimental verification**

To investigate the performance of the model in predicting the behavior of a Bingham fluid, we compared the simulation results with the analytical solutions and experimental results for the flow of a Bingham fluid between two parallel plates and dam-break type flow. The purpose of the flow test of a Bingham fluid between two parallel plates is to check whether the Bingham-Papanastasiou regularization method can correctly describe the
flow phenomena of a Bingham fluid, and the 3-D dam-break experiment is used to test the ability of the above scheme to capture the free surface of a fluid and calculate pressure and velocity.

**2.3.1. Flow between two parallel plates**

The test of flow between two parallel plates was used for the validation of the implementation of the rheological model in OpenFOAM since the analytical solution can be easily acquired from:

$$u(y) = \frac{\Delta P D}{2\mu_b L}\left(y - \frac{y^2}{D}\right) - \frac{\tau_b y}{\mu_b} \quad for\ 0 \leq y < S \tag{11}$$

Equation (11) gives the expression for velocity for $\tau > \tau_b$ (Gopala et al., 2011;Waarde, 2007), and the velocity at the plug is given by Eq. (12):

$$u_{plug} = u(S) \quad for\ S \leq y < \frac{1}{2}D \tag{12}$$

For the simulation of flow between parallel plates, a 2-D computational domain was constructed, which consisted of a rectangular channel with length $L = 2\ m$ and width $D = 0.1\ m$. The test configuration is
schematically shown in Fig. 1. The domain was discretized with $400 \times 40$ cells limited by three types of boundary faces: inlet, outlet, and no-slip walls, and the Bingham fluid flowed from the inlet to the outlet at a constant velocity of $0.069\ m/s$. Two types of Bingham fluids were simulated, namely, a self-compacting cement (SCC) mixture and grout, and their rheological parameters are shown in Table 2 with reference to Gopala (Gopala et al., 2011). In Fig. 2, the velocity profiles were compared between the numerical simulation
and the analytical solution for the SCC and the grout. The simulated results and analytical solutions appear to be in very close agreement, with the implemented results of the Bingham-Papanastasiou model fitting the analytical velocity curves very well. In terms of velocity, the average error of simulated values compared with analytical solutions is $0.000077\ m/s$, and the maximum error is $0.0016\ m/s$. For the plug flow region, the yield stress of the SCC is two orders of magnitude higher than that of the grout, so there is an obvious plug
region in the flow of the SCC, as shown in Fig. 2.

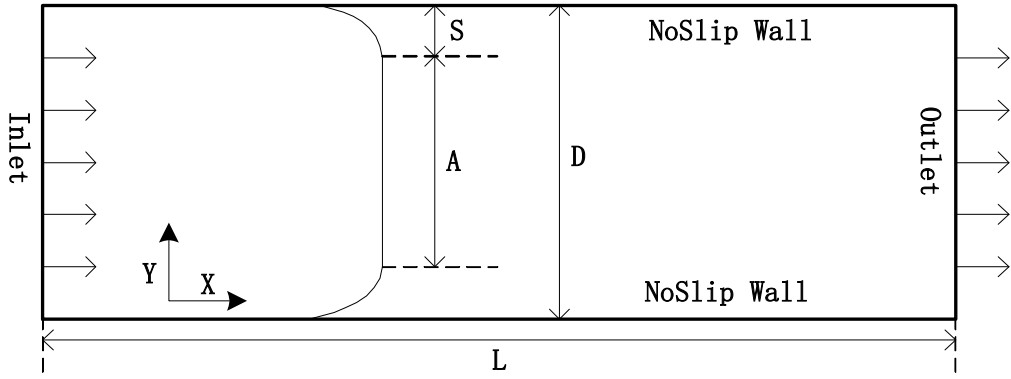

**Figure 1: Schematic of the flow between parallel plates.**

**Table 2.** Rheological parameters of a simulated Bingham fluid.

|  | **SCC** | **Grout** |
|---|---|---|
| Density (kg/$m^3$) | 2200 | 1900 |
| Yield stress (Pa) | 131 | 2 |
| Viscosity (Pa$\cdot s$) | 44.9 | 11 |


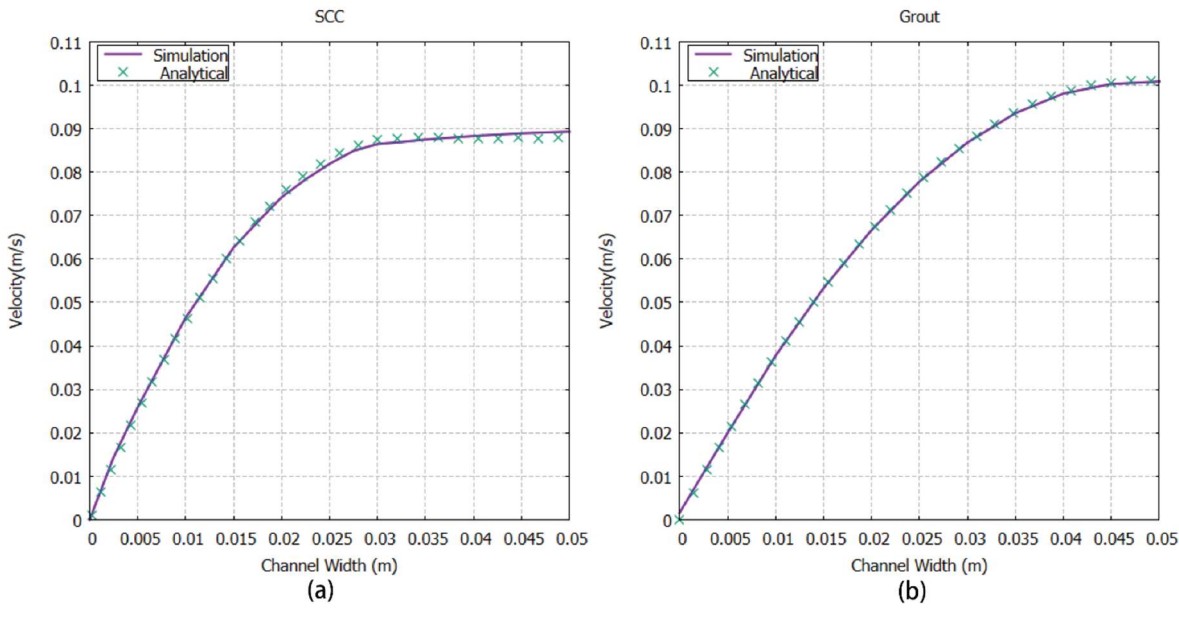

**Figure 2: Comparison of velocity variation between analytical solutions (Gopala et al., 2011) and numerical results simulated in two parallel plates based on different rheological parameters. i.e. (a) SCC and (b) Grout.**

### 2.3.2. 3-D dam-break experiment

The second experiment presented here is one of the dam-break type problems to validate the model. The dam-break problem is a classical validation case for the assessment of free surface modeling methods. As shown in Fig. 3, the experiment is carried out in a large tank of $3.22\ m \times 1\ m \times 1\ m$ with an open roof, and the right part of the tank is a rectangular box of water between a fixed wall and a temporary door. At time $t = 0\ s$, the temporary door is removed instantaneously, allowing the fluid to collapse under the influence of

gravity $g = 9.81\ m/s^2$. During the collapse, the fluid impacts an obstacle at the bottom of the tank and creates a complicated flow structure. The geometry and the probe positions are briefly described in Fig. 3;

points P1 to P8 are pressure sensors on the obstacle box, while probes H1 to H4 are used to observe the fluid heights. In this simulation, grids are used in a structural pattern with an initial spacing of $0.015\ m$, so a set of 971,498 cells are employed, and the boundary consists of the no-slip walls and the top boundary that is free to the atmosphere so that both outflow and inflow are permitted according to the internal flow. For the rheology, the typical values used in the simulations are $\mu = 0.001\ Pa\ s, \tau_b = 0\ Pa$ and $\rho = 1,000\ kg/m^3$ for water-based suspensions at $20^\circ C$.

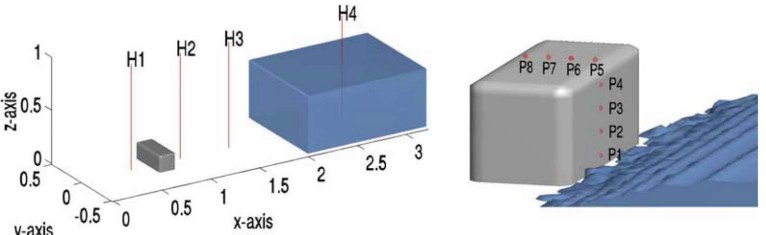

**Figure 3: Measurement positions for water heights and pressures in the dam-break experiment. (from Kleefsman (Kleefsman et al., 2005)). P1 to P8 are pressure probes, and H1 to H4 are height probes.**

The experiment was performed by the Maritime Research Institute Netherlands (MARIN), and all detailed experimental data can be downloaded free from this website (https://app.spheric-sph.org/sites/spheric/tests/test-2). In this case, we compared the simulation results against the experimental data obtained by MARIN. The free surface and velocity field of the simulation against different physical times $t = 0.4\ s$ and $t = 0.56\ s$ are shown in Fig. 4, which is similar to the results in Kleefsman (Kleefsman et al., 2005) and Lee (Lee et al., 2010). The small pictures on the top right of the screenshots represent the fluid behind the gate. As this figure suggests, the simulation results are in good agreement with the experimental data. In Fig. 5, characteristic curves of pressure and vertical water heights varying with time at P1 and H4, respectively, are depicted. Similarly, the global behaviors of the simulation and experiment are fairly consistent but slightly shifted about $0.2\ s$ after $2.55\ s$ at probe H4 and at $4.6\ s$ probe P1, as shown in Fig. 5. Kleefsman (Kleefsman et al., 2005) pointed out that an underpredicted difference could be reduced by increasing the grid resolution in the comments of their VOF simulations. At probe H4, the water height experienced a period of decline from the initial $0.55\ m$, and the experimental and simulated values reached a minimum of about $0.1\ m$ and $0.075\ m$ in approximately $0.25\ s$ and $0.27\ s$, respectively. At the pressure probe P1, the difference between the simulated pressure value and the experimental value reaches a maximum of about $1,617\ Pa$ when the pressure reaches a maximum at about $0.41\ s$, and the average error in other times is $115\ Pa$. Consequently, the ability of the above scheme to capture the free surface of a liquid and calculate the pressure and flow rate are indicated.

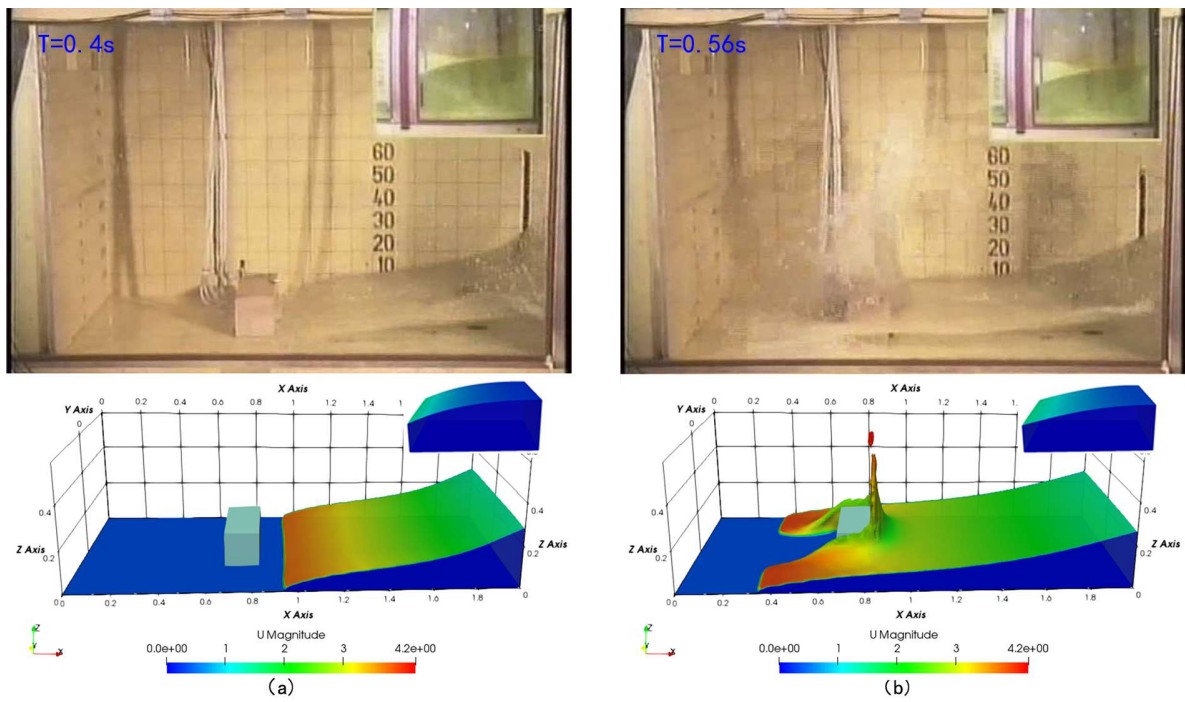

Figure 4: Screenshots of the simulation compared with the experiment (Kleefsman et al., 2005) at $t = 0.4\ s$ (a) and $t = 0.56\ s$ (b). The color map of the simulation results is mapped by velocity characteristics.

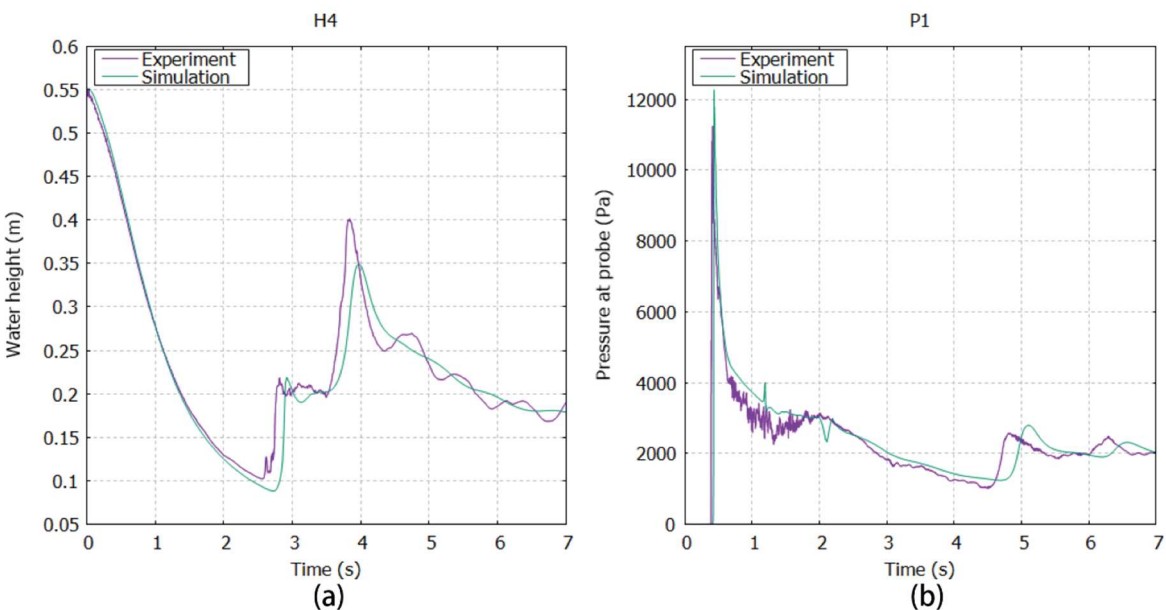

Figure 5: Comparison of vertical water heights in reservoir at probe H4 (a) and the characteristic curve of pressure changing with time at probe P1 (b) between experimental data and simulation results.

## 3. Simulation of tailings dam failures – Feijão and A'xi tailings dams

### 3.1. Feijão tailings dam

The Córrego do Feijão iron mine complex, with two tailings dams, a cargo terminal, an administrative office and a small railway network for the transport of iron ore (Fig. 6.a), is located in Brumadinho, Minas Gerais state, Brazil, in the upper valley of the Paraopeba River. The tailings dam I, measuring 86 $m$ high and 720 $m$ long, was used for disposal of approximately 12.7 million $m^3$ of iron tailings. On January 25, 2019, Dam I suffered structural damage and eventually collapsed catastrophically, causing at least 232 deaths (Santamarina et al.,

2019). According to a field survey (Porsani et al., 2019) that was conducted after the collapse, approximately 11.7 million $m^3$ of tailings were spilled from the dam, destroying office buildings, a railway bridge, and a small community. Under the action of gravity, the tailings fluids traveled 9 $km$ and eventually flowed into the Paraopeba River, the region's main river. The downstream area submerged by tailings fluids was more than 2.54 million $m^2$ based on measurements from remote sensing images from Sentinel Satellite S2 (see Fig. 6.b). According to surveillance cameras around the accident area, the velocity of the tailings fluid when the tailings dam failed was more than 10 $m/s$. Fig. 6 shows the area impacted by the Dam I collapse event.

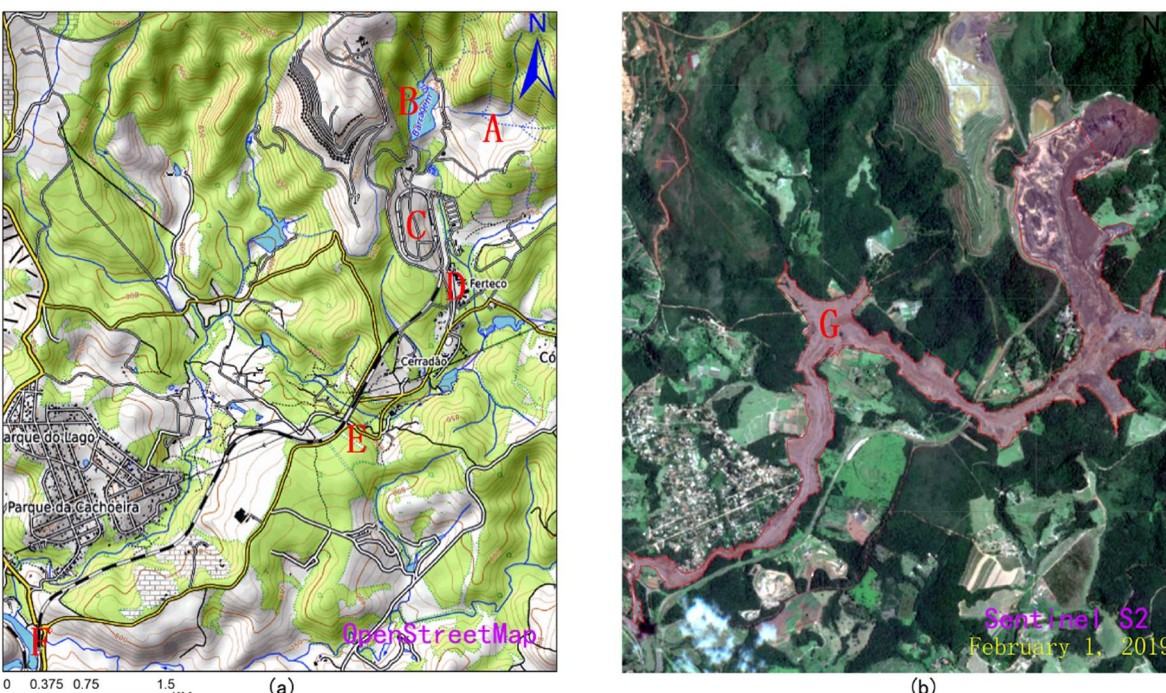

**Figure 6: Location of the Feijão iron mine complex with reconstructions of the area impacted by the 2019 Dam I failure event: (a) Location and downstream area of the Feijão tailings dam. A Location of Dam I. B Location of Dam VI which was destabilized by the event. C&E Locations of the rail network and bridge. D Location of the office center and a small community. F Location of the Paraopeba River; (b) G The destroyed downstream area. Map data: OpenStreetMap and Sentinel Satellite S2 imagery.**

### 3.2. A'xi gold tailings dam

The A'xi gold mine tailings reservoir (Fig. 7) is 58 $m$ high and located 46 $km$ from the Ili River, Xinjiang, China, in the upper valley of the A'xi River, a tributary of the Ili River. Designed for disposal of approximately 3.6 million $m^3$ of gold tailings, this tailings dam measures 210 $m$ long and occupies 170,000 $m^2$ and is the largest tailings dam in Xinjiang. The tailings storage facility of the A'xi gold mine was constructed using the 'paddock' system of construction, where the subsequent dams raised were built on the tailings. In addition, it is located on the south slope of the main ridge of Kogurqin Mountain in the Borokunu Mountains of the Western Tianshan Mountains with an elevation of 1,513 $m$ and a large downstream topographic gradient. The mining area belongs to the continental climate of the north temperate zone, where the overall rainfall is low but unusual rainfall events occur frequently. At present, the tailings dam has entered the tail period of its service life with a storage capacity of approximately 2 million $m^3$ of tailings. Additionally, the area is in an earthquake-prone zone and is prone to abnormal rainfall. Therefore, the possibility of tailings dam failure cannot be ruled out. If a

tailings accident occurs, it will cause environmental disasters and large property losses and even pollute the large reservoirs downstream. Considering the serious consequences of the A'xi gold mine tailings dam if it encounters an accident, we chose it as a disaster assessment case study to simulate and analyze its dam-break process and run-out path.

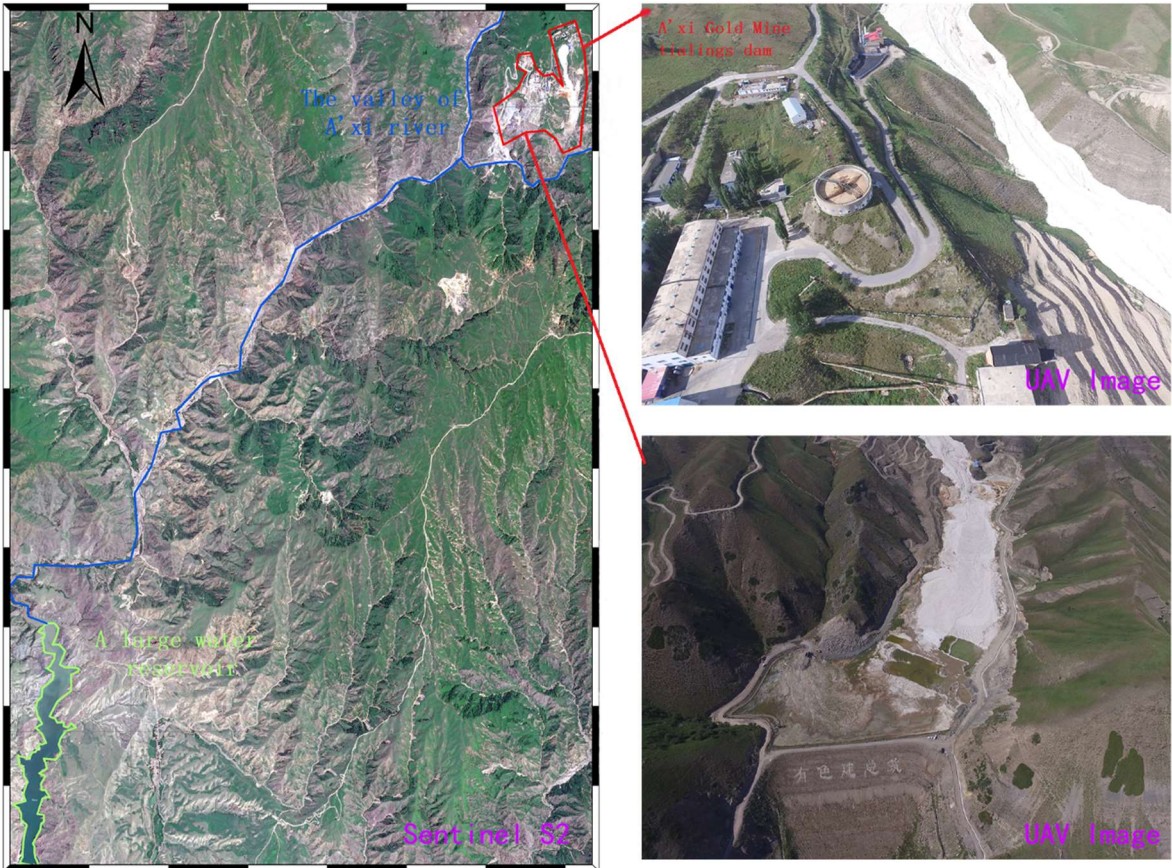

**Figure 7: Location of A'xi gold mine tailings dam: The red area is the A'xi gold mine complex; the blue line represents the valley of the A'xi River; the green area represents a large water reservoir. Image data: Sentinel Satellite S2 imagery and UAV imagery.**

### 3.3. Key factors and model setup

The presented and validated CFD model was used to simulate the run-out path and range of tailings fluid from the Feijão tailings dam failure and to simulate the breaking of the A'xi tailings dam. The features of run-out are greatly affected by three significant factors: the fineness of the downstream topography, the rheological characteristics of the tailings fluid and the initial state of the tailings fluid.

First, the terrain has a great influence on the tailings fluid's velocity and run-out path due to the gradient variations in the terrain. The fine terrain boundary, which is closer to the real landform, has more features and obvious topographic fluctuations than the coarse boundary, making the numerical simulation results more in line with the realistic phenomenon. Conversely, the coarse terrain boundary has a greater loss of detailed topographic characteristics. If the resolution of the terrain boundary is too low, it may cause a large difference between the simulation results and the real event. The topographical boundaries of a large area possibly affected by the tailings dam failure were constructed using a DEM. For the A'xi tailings dam, we went to the field to collect a group of aerial photographs with a certain overlapping range using the UAV. Based on these

overlapped images, a DEM was reconstructed with a resolution of approximately $0.5\ m\ \times 0.5\ m$, which can be further generated into a 3-D terrain model. However, due to the large altitude drop and the limited endurance of the drone used in this area, images of some areas that may be affected by tailings fluids have not been collected. For these areas, we chose the ALOS PALSAR RTC (Rdiometrically Terrain Corrected) DEM with a $12.5\ m\ \times$ $12.5\ m$ resolution released by the Alaska Satellite Facility (https://www.asf.alaska.edu/sar-data/palsar/terrain-corrected-rtc/) to replace them. Then, the UAV DEM is mosaiced into the ALOS DEM and forms a broader DEM with of cell size of $0.5\ m\ \times 0.5\ m$. After that, the inlaid DEM was georeferenced, cropped, and reconstructed to form the final 3-D terrain geometry around the A'xi Gold Mine. Because the Feijão tailings dam has collapsed, the DEM data made by drone before the accident could not be obtained; therefore, we also used the ALOS PALSAR RTC DEM to make the 3-D terrain geometry. Fig. 8 shows the DEM of the A'xi tailings reservoir and the reconstructed three-dimensional model computational grid. The 3-D model of the terrain was built to make the meshes of the computational domain, which were the data that must be used in numerical calculations based on the FVM. For mesh generation, an OpenFOAM utility, snappyHexMesh, was employed to generate 3-D meshes containing hexahedra and split-hexahedra automatically from a 3-D file (STL or OBJ format).

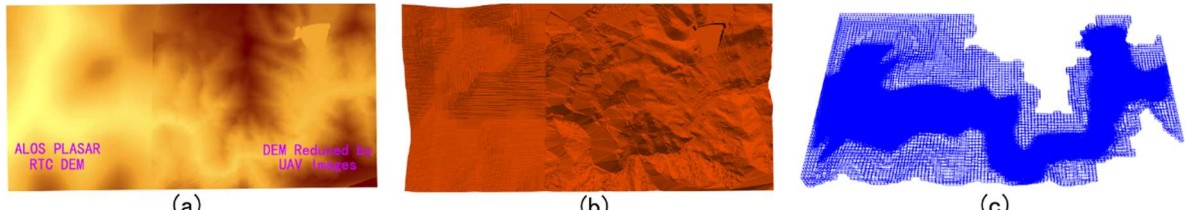

**Figure 8: The 3-D terrain model around the A'xi Gold Mine. (a) the fine DEM generated by UAV (right) and ALOS satellite (left), (b) reconstructed 3-D terrain model and (c) the computational meshes.**

Furthermore, the travel distance and extent of a tailings outflow are affected by the rheological equation. Several studies (Pastor et al., 2004;Henriquez and Simms, 2009;Gao and Fourie, 2015;Babaoglu and Simms, 2017;Gao and Fourie, 2019) have established that Bingham constitutive law provide a reasonably good compromise between accuracy and simplicity, and it has been frequently adopted to describe the rheology of tailing fluid. Respect for the determination of the rheological parameters, Henriquez and Simms (Henriquez and Simms, 2009) determined the yield stress and viscosity of tailings flow using rheometer and slump tests, the results showed that yield stress ranges roughly from $40\ Pa$ to $300\ Pa$ and dynamic viscosity ranges from approximately $0.1-1.12\ Pa\ s$ when the solid concentration is between 60% and 70%. Gao and Fourie (Gao and Fourie, 2015) concluded that the range of yield stress values is from approximately $18\ Pa$ to $60\ Pa$ which covers the majority of cases of actual tailings disposal operations, with a viscosity ranging from around $0.3-0.8\ Pa\ s$. The yield stress of the tailings flow generally has more influence on the final profile than does velocity during the deposition process in flume tests, while the importance of viscosity for the formation of the final profile of tailings increases with a rise of inertia effects (Gao and Fourie, 2019). In this study, the Bingham viscosity and yield stress were determined based on a previous investigation report conducted in testing the rheological properties of tailings (Liao and Zhou, 2015). Liao (Liao and Zhou, 2015) collected tailings samples from three different tailings ponds and mixed these tailings with water to form different concentrations of tailings fluid for rheological tests. A total of 15 rheological tests were carried out using LAMY RM100

rotational viscometer, with a range of dams (#1-#3), and volume concentrations of mixtures of 60%-80%. The #1 tailings sample and #2 tailings sample were mainly fine sand with a max particle size of less than 2 mm, but the powder content (0.075 mm < diameter < 0.25mm) of #1 tailings sand (68.68%) was more than that of #2 tailings sand (42.95%). The tailing #3 tailings sample was mainly composed of coarse sand with a max particle size of less than 5 mm. Because Feijão iron tailings and A'xi tailings are similar to # 1 tailings, they are both silt and have a small particle size, and the main mineral composition of # 1 tailings is iron ore. In this work, the 80% solid concentration of the #1 tailings dam is selected, corresponding to a density 1,830 $kg/m^3$, a viscosity of 0.741 $Pa\,s$ and yield stress of 59.82 $Pa$. The values of this set of parameters are in accordance with the recommended viscosity and yield stress range of Henriquez (Henriquez and Simms, 2009) and Gao (Gao and Fourie, 2015) et al.

Undoubtedly, the initial conditions of the released tailings fluid can have an important effect on the velocity and out-flow distance of the tailings, especially at the time of the dam break, which is mainly caused by the initial geometry and volume of the tailings. For the Feijão tailings dam, according to the monitoring video of the accident area, the structure of the dam body completely collapsed instantaneously, and the same hypothesis was also used for the A'xi tailings pond. The volume of tailings released by the Feijão tailings accident was set to 11.7 million $m^3$ based on subsequent reports. While the Feijão tailings dam almost discharged all the tailings it had stored, tailings dam accidents will generally only release part of the tailings stored, according to many historical accidents. The volume of tailings released after the dam break at the A'xi Gold Mine tailings reservoir is estimated to be approximately 756,000 $m^3$ based on the regression equation proposed by Larrauri (Larrauri and Lall, 2018):

$$\log(V_F) = -0.477 + 0.954 \times \log(V_T) \tag{13}$$

where $V_T$ is the total impounded volume; $V_F$ is the volume of tailings that could potentially be released. For the initial 3-D model of the tailings dam, its volume is equal to the total volume of tailings released, which is constructed based on the dam height and geometric shape of the tailings pond. Formerly, the 3-D topographic model was modified to suit the initial geometry of the tailings.

Generally, a smaller cell spacing provides a better simulation result, but it will dramatically increase the time for calculation. In our simulation, the cell spacing of the Feijão tailings dam and the A'xi tailings dam are fixed to $10\,m \times 10\,m \times 3\,m$ and $3\,m \times 3\,m \times 3\,m$, generating 3,241,998 cells and 6,657,418 cells, respectively. The terrain elevation within each cell is evenly distributed, so it is necessary to supplement relatively accurate surface information such as terrain roughness. Owing to the difference in the surface cover, the roughness length of the downstream terrain of the Feijão tailings dam and the A'xi tailings dam are set to $1\,m$ and $0.1\,m$, respectively, based on a criterion given by Henderson-Sellers (Henderson-Sellers et al., 1993). Finally, the geometry and associated fields are decomposed into pieces using the DecomposePar utility, and each separate part of the decomposed domain was run on the cluster of the Fujian HPC by using the IntelMPI (an implementation of the standard message passing interface) and SLURM workload manager.

### 3.4. Results and discussion

The proposed 3-D CFD approach was used to simulate the dynamic behavior of tailings flow over the 3-D topography based on the above settings, and the discussion of the propagation features of the tailings outflow is presented as follows.

    For the Feijão tailings reservoir, we simulated the travel of tailings for $2,500 \, s$ at the time of the disaster.
The typical simulation snapshots of the fluidization process of the 2019 Feijão dam-break event are presented in Fig. 9. In addition, to analyze the movement characteristics of the tailings after failure, the progress of the front displacement within time sequence is shown in Fig. 10. After dam failure, the gravitational potential energy of the tailings fluid was converted into kinetic energy, causing the velocity to increase suddenly, even more than 20 $m/s$. Subsequently, it took 150 $s$ for the tailings fluid to travel approximately $1,300 \, m$ and
reach the office center and small community, with a velocity of 15 $m/s$. After 150 $s$, the front flow smashed the railway bridge at approximately $3,150 \, m$ downstream, with a decreasing average velocity of less than 5 $m/s$. Finally, it flowed into the Paraopeba River at approximately $2,400 \, s$, with a small average velocity of less than 2 $m/s$, and the total travel distance was approximately 9 $km$. Fig. 11 shows the mean velocity time-history of the free surface. It can be obtained that the average velocity of the tailings flow increases
continuously to more than 6 $m/s$ in the first $100 \, s$, and then it gradually slows down, and the average speed stabilizes at around 1 $m/s$ from $1,000 \, s$. Furthermore, to check the quality of the results simulated on the Feijão event, a comparison of the CFD simulated and measured submerging area is shown in Fig. 12, from which we can see that the numerical simulation results of the tailings flow directions and impact area are also modeled satisfactorily. There is a significant difference in the location of the railway network (see Fig. 6.a and
12), which is mainly due to the acquisition time of the DEM used in 2011 when the railway network had not been built. The simulated inundation area is about 2.572 $km^2$, which is only 0.036 $km^2$ different from the actual measured inundation area of 2.536 $km^2$. It can be observed that the simulation results are fairly consistent with the on-site historical run-off area, with only minor differences in some locations, and we believe that more accurate DEM data and smaller grids' size could be helpful to obtain a better result.
However, the collapse of the tailings pond is a very complicated geographical phenomenon. It is impossible to fully satisfy the historical situation in the simulation because uncertainty factors are not considered, so it might not be closely matched in terms of time.

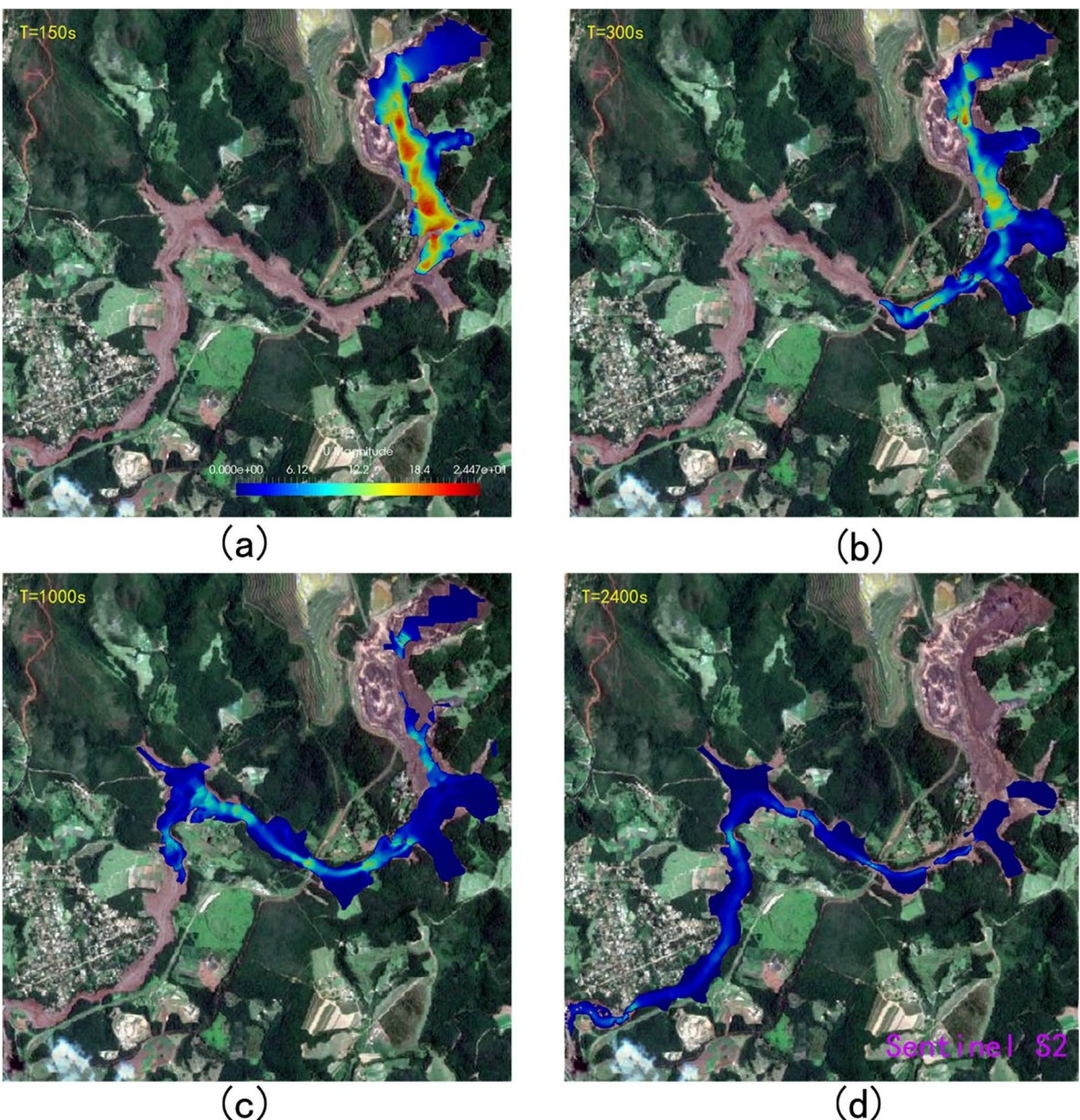

**Figure 9: The time sequence of the numerically computed event up to the tailings reaching the Paraopeba River. The base map is the satellite image (Sentinel S2) after the dam break. The color map of the simulation results is mapped by velocity characteristics.**

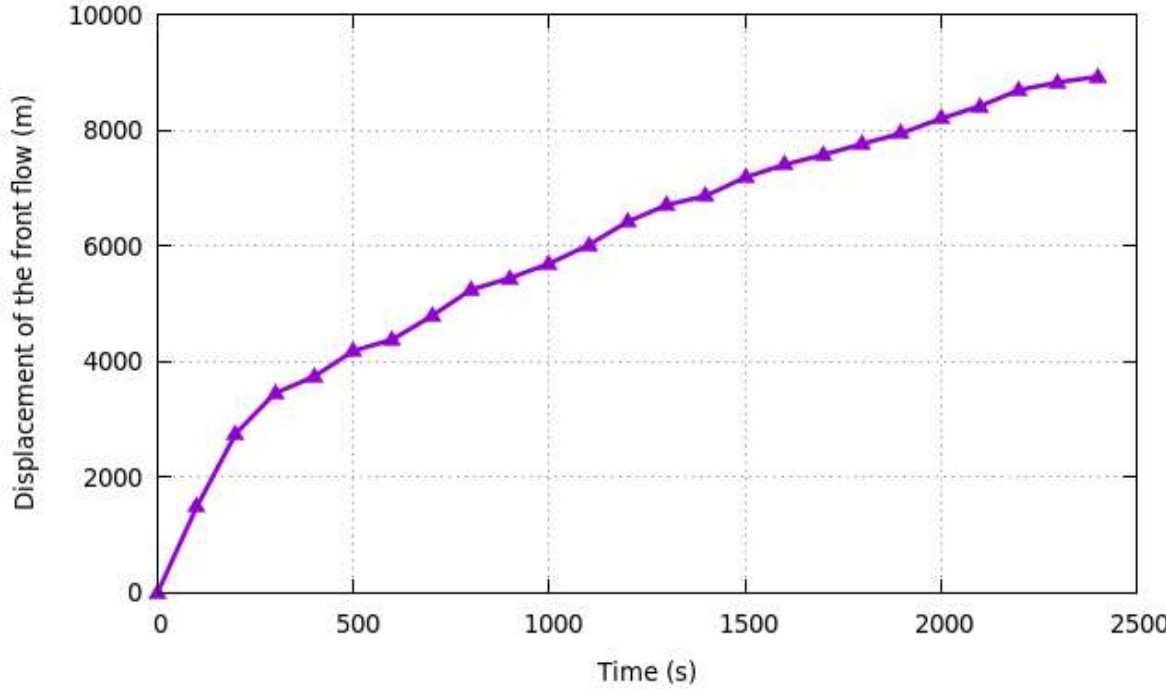

Figure 10: Displacement time-history for the front of Feijão failure event in the simulation.

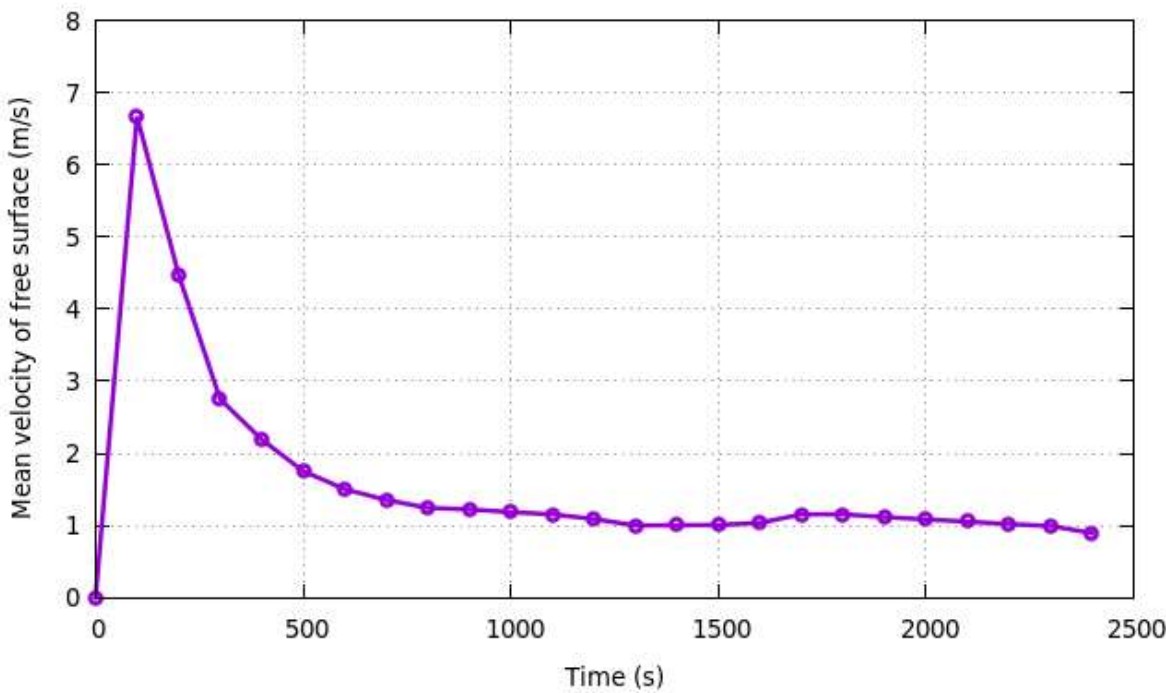

Figure 11: Average velocity time-history for the free surface simulated of the Feijão failure event.

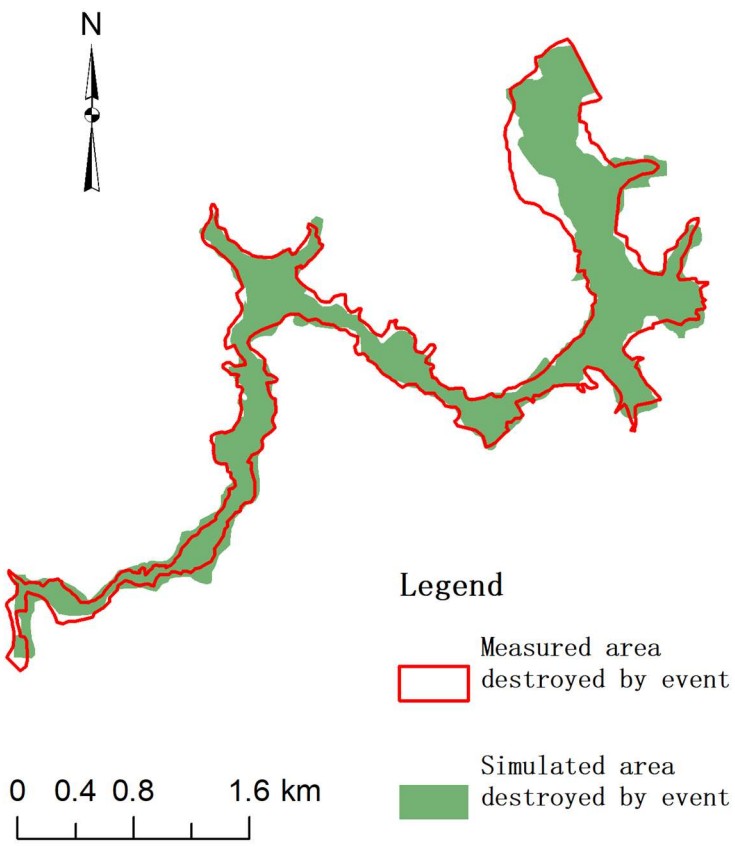

 **Figure 12: Comparison of simulation results and measurement results in the area destroyed by the Feijão accident.**

According to the relevant settings in Section 3.3, if the A'xi gold mine tailings pond experiences a dam failure, the flow path and submerged range of the tailings were predicted within 800 $s$. The simulated propagation of the tailings fluid released from the A'xi tailings dam failure is shown in Fig. 13. In addition, to analyze the movement characteristics of the tailings after failure, the progress of the front displacement within time is shown in Fig. 14. After the tailings fluid is released from the breach, it flows down along the terrain under the action of gravity. At 20 $s$, it first submerges the environmental protection depot and power distribution room (see Fig. 13) approximately 250 m downstream of the tailings pond, and the speed exceeds 15 $m/s$. Immediately afterward, the tailings fluid enters the A'xi Valley, approximately 320 $m$ from the dam site, and flows downstream along the terrain in the valley. Although the elevation difference in the valley is more than 155 $m$ (in the calculation domain), the valley is so tortuous that the flow rate of the tailings fluid gradually decreases. Eventually, at 760 s, the front reaches the simulated boundary with an average velocity of less than 2 $m/s$ and a flow distance of approximately 3.5 $km$. Fig. 15 shows the mean velocity time-history of the free surface. It can be obtained that the average velocity of the tailings flow increases continuously to more than 11 $m/s$ along the steep channel in the first 10 $s$, and then it gradually slows down due to the continuous curved valley, and the average speed stabilizes at around 2 $m/s$ from 200 $s$ to 550 $s$. After that, there is a slight increase in speed due to the large topographic elevation changes. It can be seen from the numerical results that almost 75,600 $m^3$ of tailings released after the break of the A'xi gold mine tailings dam are almost poured into the A'xi Valley except that some of the tailings are blocked by the environmental protection depot. The normal

runoff volume of the A'xi River is 0.5-2 $m^3/s$, and the runoff volume during the dry season is 0.002 $m^3/s$. Overall, the runoff volume is relatively small, so the influence of water flow is not considered in the simulation. However, the region is prone to abnormal rainfall events, which lead to an increase in river runoff. Under this situation, the toxic tailings that flow into the river valley would be transported by the river to the large reservoir downstream, or even eventually to the Ili River, causing serious enviromental pollution.

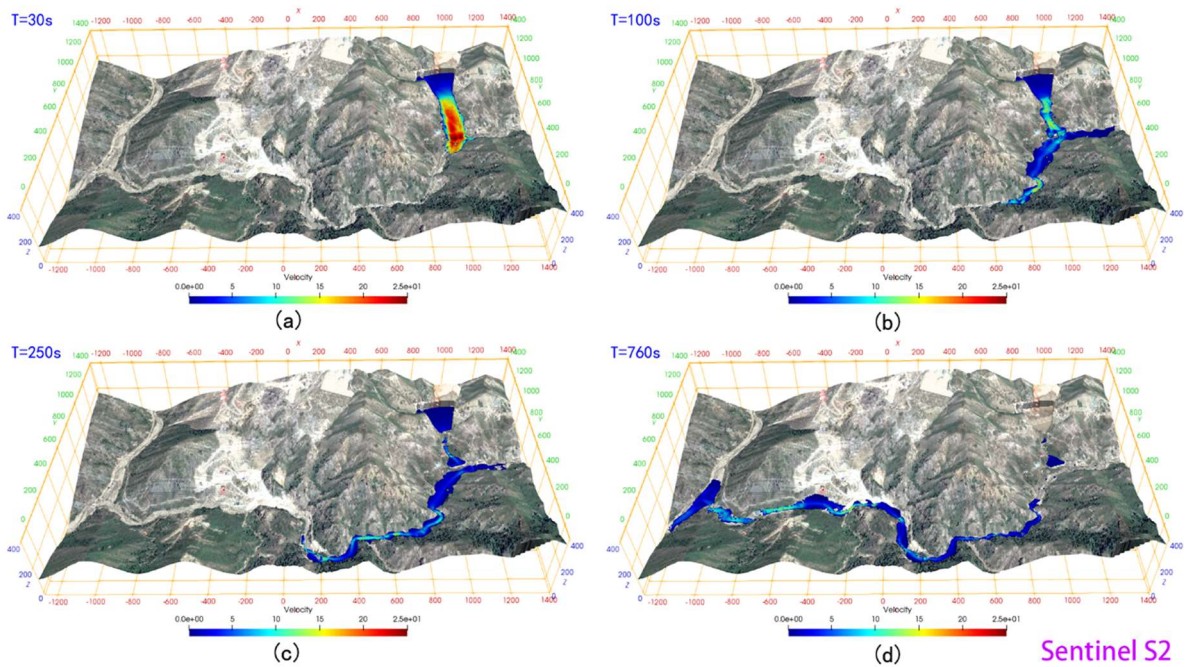


**Figure 13: The time sequence of the numerically computed event up to the tailings reaching the left boundary of the computational domain. The base map is the satellite image (Sentinel S2). The color map of the simulation results is mapped by velocity characteristics.**

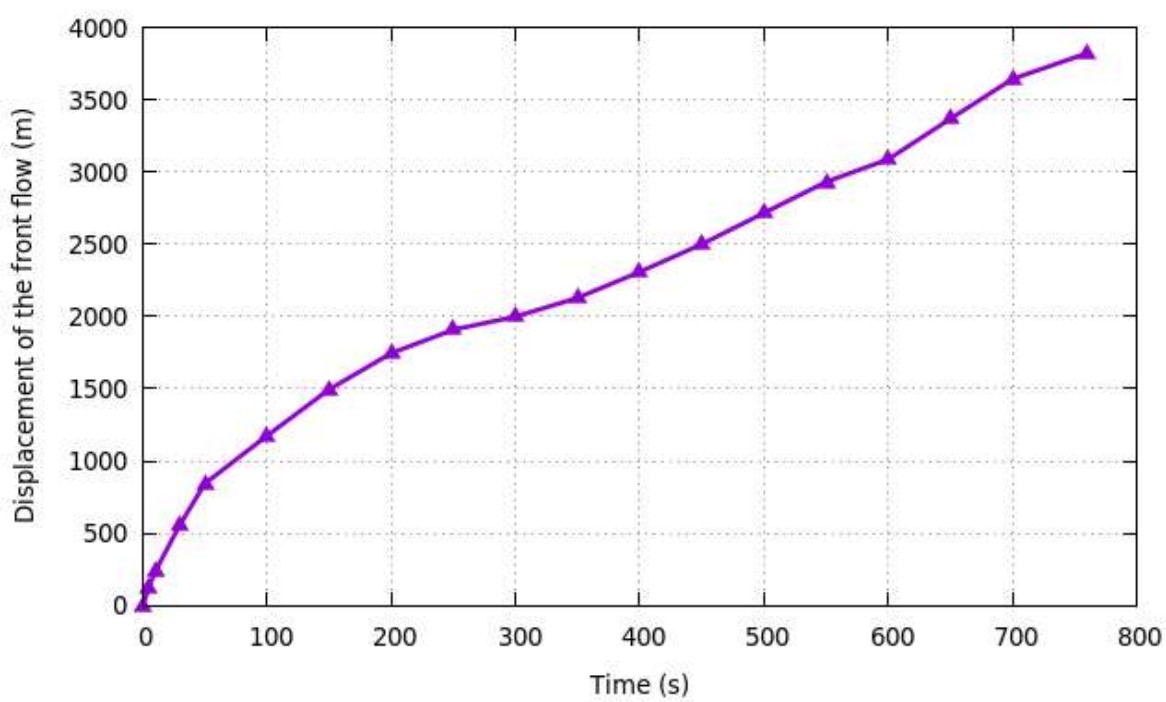


**Figure 14: Displacement time-history for the front of A'xi failure in the simulation.**

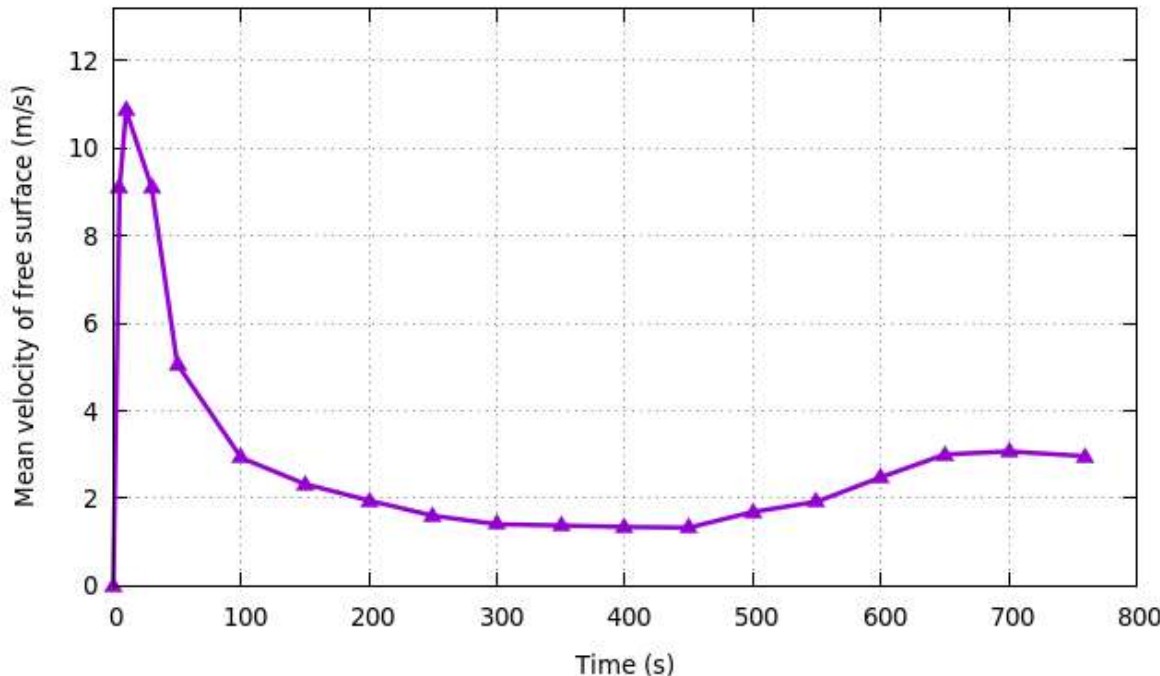

**Figure 15: Average velocity time-history for the free surface simulated of the A'xi dam failure.**


From the simulated results shown above, it can reasonably be concluded that the proposed method can accurately represent the whole dynamic processes of tailings movement after dam failure. In addition, parallel technology was used to speed up computation. We tested the parallel speedup for different numbers of processors on the supercomputing cluster of the Fujian HPC, with a computational load of more than 3 million cells, and the speedup

was nearly 50 times. When the number of processors was more than 50, the calculation time was no longer reduced because the acceleration ratio also depends on the amount of computation. If the number of cells increases, the speedup will be greater. Fig. 16 shows the parallel speedup of different numbers of processors.

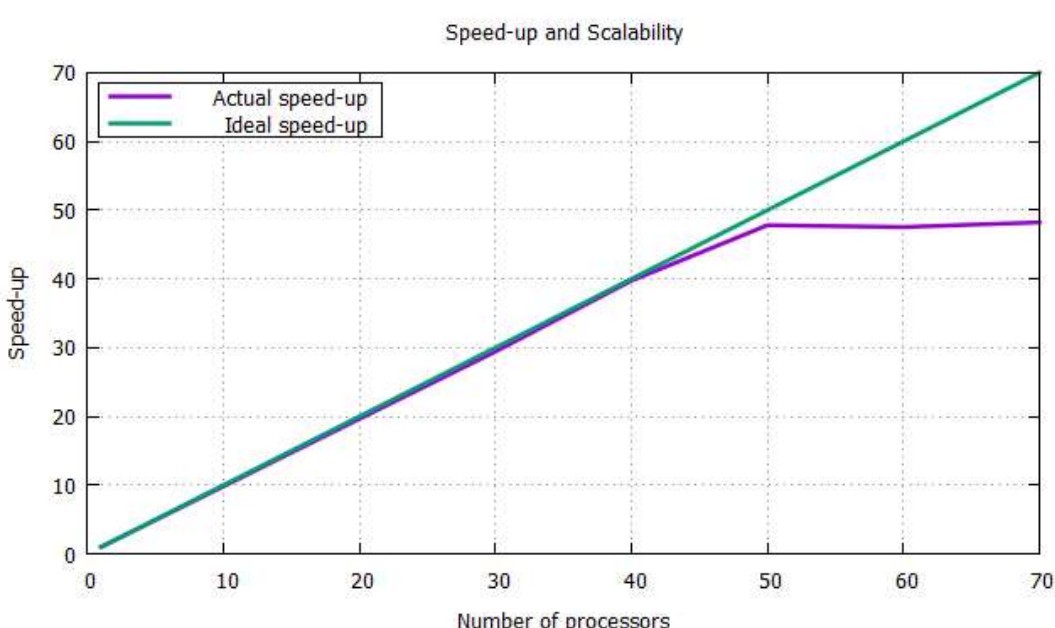


**Figure 16: Parallel speedup with different numbers of processors. The green line is the ideal speedup, and the purple line is the actual speedup.**

## 4. Conclusions

The damage caused by tailings dam failures occurring frequently around the globe can be costly and
catastrophic. The topic of disaster prevention and mitigation is being discussed more in government departments and the mining industry. The debris flow from a tailings dam failure is a very complex phenomenon involving physical mechanisms and the evolution of fluid properties. In this paper, a 3-D CFD model based on the FVM was proposed to predict the routing and impact area of tailings across complex topography during tailings dam failure. The Navier–Stokes equations and the Bingham rheology model are
introduced into the FVM framework as the governing equations and the constitutive model, respectively. To construct a wide range of detailed downstream terrain, UAV photogrammetry, and satellite imagery are used to sample surface information. Moreover, parallel technology was used to simulate the complex problem of the supercomputing cluster to achieve rapid and reasonable predictions for disaster prevention and mitigation.

To verify the performance of the reported scheme, the numerical model is first tested against an analytical solution in the literature for the flow of a Bingham fluid between two parallel plates, demonstrating the correct implementation of the Bingham-Papanastasiou constitutive model in OpenFOAM. Next, the numerical model is validated against the experimental value for the 3-D dam-break experiment, thereby ensuring a good prediction of the free surface profile. To further illustrate the performance of the presented method, the 2019
Feijão dam-break event in Brazil was selected as a case study, and the numerical simulation results coincide well with the in situ investigation. Finally, considering the serious consequences of the A'xi gold mine tailings dam if it encounters an accident, we chose it as a disaster assessment case study to simulate and analyze its dam-break process and run-out path. Therefore, CFD numerical modeling can provide an effective means for mapping hazardous areas, estimating hazard intensity, and designing appropriate protective measures.

**Code availability:** Code can be made available by the authors upon request.

**Data availability:** Data can be made available by the authors upon request.

**Author contributions:** LT, DY, and CC conceived and designed the method; DY carried out the simulations, produced the results, and wrote the original manuscript under the supervision of LT. LT and CC writing-review and editing.

**Competing interest:** The authors declare that they have no conflicts of interest.

**Acknowledgments:** The authors would like to thank Le Wang, Ph.D. from Xi'an Shiyou University for his expertise in developing the CFD. The numerical calculations in this paper have been performed on the supercomputing system in the Supercomputing Center of Fujian. The authors acknowledge the OpenFOAM Foundation, which developed the open-source CFD code and released it to the public.

**Financial support:** This research was funded by the National Key Research and Development Program of China (grant no.
2017YFB0504203).

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
