# Peer review of "Three-dimensional numerical simulation of mud flow from a tailings dam failure across complex terrain"

_Natural Hazards and Earth System Sciences, 2019_

## Referee Comment (RC1) · Anonymous Referee #1 · 21 Oct 2019

The paper discusses a 3D computational fluid dynamics approach for predicting the flow routing and impact area of mud flow from a dam failure. It is a very useful idea for dealing with the tailings dam accidents that can cause serious disasters. But there are still some details that should be considered firstly. My comments and suggestions are the following.

1. Three-dimension (3D) technology can benefit to the simulation of the real world. And in this paper, it is used for the simulation of mud flow. However, this paper seems not giving a satisfying overview on the use of 3D technology in the related fields of tailings dam accident.

[Figure]

2. The author used much text to introduce the numerical models used in this paper, however, the improvement of these models for dealing with tailings dam accident doesn't seem to be described clearly.

3. During the simulation for the A'xi tailings dam, the author used two different DEMs with 0.5m × 0.5m and 12.5m × 12.5m resolutions. Will the boundary inconsistency of two areas with different DEMs cause the low accuracy of simulation? How did the author address the inconsistency while reconstructing the 3D terrain?

4. The parameters of models is usually important. In this paper, how did the authors obtain the parameters for the simulations of different tailings dams?

5. How to evaluate the simulation results, is there some quantitative methods was used for result evaluation?

---

## Referee Comment (RC2) · Anonymous Referee #2 · 31 Oct 2019

Manuscript Id: nhess-2019-298

Title: Three-dimensional numerical simulation of mud flow from a tailings dam failure across complex terrain Authors: Dayu Yu, Liyu Tang, and Chongcheng Chen

General comments: The submitted manuscript describes a 3D method to model a mud flow event from a dam failure using OpenFOAM source code. The authors model the flow using the Bingham-Papanastasiou constitutive law and analyse the model performance simulating the obtained results by means of an experimental verification. After that, the authors use the model to simulate two scenarios of tailings dam failures. Simulation results obtained are poorly analysed and describe only in qualitatively way, while

a quantitative analysis should be provided. The presented paper could provide a good addition to the literature, showing a good fit with the journal main topics, but only after accurate revisions.

Main suggestions are: The figures are generally not clear and too small relating to the information contained. There is a poorly coherence with the text content, especially in figure 2 and 3. A quantitative analysis of the simulation results is missing in the experimental verification and in both dam failures simulations. Historical results are cited to confirm the goodness of the simulation, but no field data or observations are presented to validate the goodness of the modelling procedure. It is not clear the cell size used in the terrain model and which DEM is used for the A'xi tailing dam event (pre or post event). The authors attributed some rheologic parameters to the modelled flows without performing a back analysis to define such parameters and to justify their choice. It would be better to specify in the paper that this aspect has to be take into account in the simulation of a real case.

---

## Author Comment (AC1) · 22 Nov 2019

al.**

**Dayu Yu et al.**

13070895895@163.com

Dear Referee, thank a lot for your appreciation of our paper and for the work you did on our manuscript. We greatly appreciate your valuable comments as they may contribute to increase the manuscript robustness and, in general, to improve its quality and readability. In the following, we supply a point by point answer to the general and specific comments raised by the referee (see also attached file).

General comment: Simulation results obtained are poorly analysed and describe only in qualitatively way, while a quantitative analysis should be provided.

[Figure]

Answer: We thank you for this comment. In the revised of the manuscript, we have added some quantitative analysis and describe in the experimental verification and in both dam failures simulations according this comment on page 7, page 9 and page 16-21 (marked in blue).

Comment 2: The figures are generally not clear and too small relating to the information contained. There is a poorly coherence with the text content, especially in figure 2 and 3.

Answer: Thank you for your comment. We have made the corresponding revision according to your comments in the revised manuscript. The specific modifications are as follows: we replaced Fig. 5 and 16 in the previous manuscript with clearer figures; We revised the captions of Fig. 2 and 3 to be consistent with text content in the revised manuscript (marked in blue).

Comment 3: A quantitative analysis of the simulation results is missing in the experimental verification and in both dam failures simulations.

Answer: We thank you for this comment. In the revised of the manuscript, we have added some quantitative analysis and describe in the experimental verification and in both dam failures simulations according this comment on page 7, page 9 and page 16-21 (marked in blue).

Comment 4: Historical results are cited to confirm the goodness of the simulation, but no field data or observations are presented to validate the goodness of the modelling procedure.

Answer: Indeed, there is lack of field data or observations in situ investigation in evaluating the simulation results of real case. Because the occurrence of tailings dam break is uncontrollable and some quantitative information in the process of dam break, it is difficult to collect the data of occurrence process when collapsing. In the revised manuscript, Fig.6 shows satellite images obtained after the Feijiao DamâĚăcollapse,

which recorded destroyed downstream area. Consequently, we measured the submerging area and total travel distance of tailings fluid, and made comparation to the simulation results. Fig.12 showed that simulated and measured submerging area are in agreement.

Comment 5: It is not clear the cell size used in the terrain model and which DEM is used for the A'xi tailing dam event (pre or post event)?

Answer: During the simulation for the A'xi tailings dam, we used two different DEMs (UAV DEM and ALOS DEM) with 0.5m × 0.5m and 12.5m × 12.5m resolutions, respectively. The UAV DEM is mosaiced into the ALOS DEM and forms a final DEM with of cell size of 0.5m × 0.5m resolution. But, the effective precision in the extent of ALOS DEM is still only 12.5m × 12.5m. We have supplied the relevant description on Page 14 of the revised paper.

Comment 6: The authors attributed some rheologic parameters to the modelled flows without performing a back analysis to define such parameters and to justify their choice.

Answer: To justify our choice in rheological parameters, some discussions on the value of rheological parameters are supplemented on page14-15 of the revised version. In this study, the Bingham viscosity and yield stress were determined based on a previous investigation report conducted in testing the rheological properties of tailings using LAMY RM100 rotational viscometer (Liao and Zhou, 2015). Liao (2015) collected several tailings samples and conducted 15 sets of tailings fluid rheology tests, and the test operation complies with the rheological test specification. We selected the test results of one group of the iron ore tailings (Feijiao tailings dam is also an iron ore tailings pond) as rheological parameters for the test.

Please also note the supplement to this comment:
https://www.nat-hazards-earth-syst-sci-discuss.net/nhess-2019-298/nhess-2019-298-AC1-supplement.pdf

**Supplement:**

[revised manuscript text omitted]

---

## Author Comment (AC2) · 22 Nov 2019

Dear Referee, thank a lot for your valuable and helpful comments concerning our manuscript. Please note that the revised manuscript has been attached in supplement file. The main corrections in the manuscript and the point-by-point responses to your comments are as following:

Comment 1: Three-dimension (3D) technology can benefit to the simulation of the real world. And in this paper, it is used for the simulation of mud flow. However, this paper seems not giving a satisfying overview on the use of 3D technology in the related fields of tailings dam accident.

[Figure]

Answer: It is a novel method to predict the flow range of tailings fluid on real 3-D terrain using 3-D CFD approach. Most prediction of flow range of tailings fluid was implemented using 2-D models. In the introduction, we revised the overview and supplemented the 3-D method for the study of debris flow or landslides on page 2 in lines 60-106.

Comment 2: The author used much text to introduce the numerical models used in this paper, however, the improvement of these models for dealing with tailings dam accident doesn't seem to be described clearly.

Answer: We have made the corresponding revision according to your comments on page 6, lines 193-196. Since the rheological properties of mudslide fluids and tailings fluids are similar we used a series of CFD models or methods to solve run-out problem for predicting the inundation area of tailings fluid after tailings pond dam failure. To make it possible for the tailings-flow simulation, we integrated the Bingham-Papanastasiou model into the original code of OpenFOAM.

Comment 3: During the simulation for the A'xi tailings dam, the author used two different DEMs with 0.5m × 0.5m and 12.5m × 12.5m resolutions. Will the boundary inconsistency of two areas with different DEMs cause the low accuracy of simulation? How did the author address the inconsistency while reconstructing the 3D terrain?

Answer: Just as your keen-insight comment, there is an inconsistency problem that is common in the mosaic process of different resolution DEM. We have done some processing on the original DEMs, and after processing, the inconsistency problem is greatly alleviated. The process is as follows: First, the two DEMs with different resolutions were georeferenced to ensure a geographical match. Then we compared the elevation values of many identical feature points of the two DEM, and found that the elevation values of UAV DEM are generally about 40 m higher than those of ALOS. All pixel values of the UAV DEM are resampled by 40 meters to reduce the elevation difference of the two DEM at the boundary line. Finally, two DEMs are mosaiced together in

the geographic information system, such as ArcGIS and QGIS. This is not the perfect way to deal with the inconsistency of the boundary, and there is still some error at the joint, but we think that this local error will not significantly affect the simulation results, which is acceptable.

Comment 4: The parameters of models is usually important. In this paper, how did the authors obtain the parameters for the simulations of different tailings dams?

Answer: In fact, for the rheological parameters of tailings, we selected a group of rheological test data (Section 3.3) of tailings similar to that of simulated tailings in a series of rheological tests conducted by Liao et al.

Comment 5: How to evaluate the simulation results, is there some quantitative methods was used for result evaluation?

Answer: For the evaluation of the simulation results, we first simulated an analytical verification test and a laboratory validation test to illustrate that this method was used to simulate tailings fluids by some quantitative comparison. Then, we simulate a real case of tailings dam break ïïjĹFeijiao tailings dam in BrizialïïjĽ, and the simulated routing and destroyed range coincided well with satellite images obtained after the Feijiao DamâĚăcollapse, which recorded destroyed downstream area, but there is lack of other field data or observations in evaluating the simulation results of real case because the occurrence of tailings dam break is uncontrollable and some quantitative information in the process of dam break is difficult to collect.

Please also note the supplement to this comment:
https://www.nat-hazards-earth-syst-sci-discuss.net/nhess-2019-298/nhess-2019-298-AC2-supplement.pdf

**Supplement:**

[revised manuscript text omitted]

---

## Author Response (AR1)

Dear editor,

Thank a lot for the work you did on our manuscript. We thank the Referees #1 and #2 for their thorough work to improve our manuscript and useful comments and suggestions. We took into account all of them and revised the manuscript accordingly, and the revised content has been marked in red in the revised manuscript. The detailed point-by-point response is given below:

Referee #1

**Comment 1:** Three-dimension (3D) technology can benefit to the simulation of the real world. And in this paper, it is used for the simulation of mud flow. However, this paper seems not giving a satisfying overview on the use of 3D technology in the related fields of tailings dam accident.

**Response:** It is a novel method to predict the flow range of tailings fluid on real 3D terrain using 3-D CFD approach. Most prediction of flow range of tailings fluid was implemented using 2-D models. In the introduction, we revised the overview and supplemented the 3-D method for the study of debris flow or landslides on page 2 in lines 60-106..

**Comment 2:** The author used much text to introduce the numerical models used in this paper, however, the improvement of these models for dealing with tailings dam accident doesn't seem to be described clearly.

**Response:** We have made the corresponding revision according to your comments on page 6, line 193. Since the rheological properties of mudslide fluids and tailings fluids are similar we used a series of CFD models or methods to solve run-out problem for predicting the inundation range of tailings fluid after tailings pond dam failure. To make it possible for the tailings-flow simulation, we integrated the Bingham-Papanastasiou model into the original code of OpenFOAM..

**Comment 3:** During the simulation for the A'xi tailings dam, the author used two different DEMs with 0.5m × 0.5m and 12.5m × 12.5m resolutions. Will the boundary inconsistency of two areas with different DEMs cause the low accuracy of simulation? How did the author address the inconsistency while reconstructing the 3D terrain?

**Response:** Just as your keen-insight comment, there is an inconsistency problem that is common in the mosaic process of different resolution DEM. We have done some processing on the original DEMs, and after processing, the inconsistency problem is greatly alleviated. The process is as follows: First, the two DEMs with different resolutions were georeferenced to ensure a geographical match. Then we compared the elevation values of many identical feature points of the two DEM, and found that the elevation values of UAV DEM are generally about 40 m higher than those of ALOS. All pixel values of the UAV DEM are resampled by 40 meters to reduce the elevation difference of the two DEM at the boundary line. Finally, two DEMs are mosaiced together in the geographic information system, such as ArcGIS and QGIS.

This is not the perfect way to deal with the inconsistency of the boundary, and there is still some error at the joint, but we think that this local error will not significantly affect the simulation results, which is acceptable.

**Comment 4:** The parameters of models is usually important. In this paper, how did the authors obtain the parameters for the simulations of different tailings dams?

**Response:** In fact, for the rheological parameters of tailings, we selected a group of rheological test data (Section 3.3) of tailings similar to that of simulated tailings in a series of rheological tests conducted by Liao et al.

**Comment 5:** How to evaluate the simulation results, is there some quantitative methods was used for result evaluation?

**Response:** For the evaluation of the simulation results, we first simulated an analytical verification test and a laboratory validation test to illustrate that this method was used to simulate tailings fluids by some quantitative comparison. Then, we simulate a real case of tailings dam break (Feijiao tailings dam in Brizial), and the simulated routing coincided well with the in situ investigation, but there is no quantitative method in evaluating the simulation results of real case because the occurrence of tailings dam break is uncontrollable and some quantitative information in the process of dam break is difficult to collect

Referee #2

**General comment:** Simulation results obtained are poorly analysed and describe only in qualitatively way, while a quantitative analysis should be provided.

**Response:** We thank you for this comment and we agree with you on this point. Indeed, the simulation results is insufficient in quantitative analysis in our manuscript. In the revised of the manuscript, we have added some quantitative analysis and describe in the experimental verification and in both dam failures simulations according this comment on page 7, page 9 and page 16-21 (marked in blue).

**Comment 2:** The figures are generally not clear and too small relating to the information contained. There is a poorly coherence with the text content, especially in figure 2 and 3.

**Response:** Thank you for your comment. We have made the corresponding revision according to your comments in the revised paper. The specific modifications are as follows: we replaced Fig. 2, 5 and 16 in the previous manuscript with clearer figures; We revised the captions of Fig. 2 and 3 to be consistent with text content in the revised manuscript (marked in blue).

**Comment 3:** A quantitative analysis of the simulation results is missing in the experimental verification and in both dam failures simulations.

**Response:** We thank you for this comment. Indeed, the simulation results is insufficient in quantitative analysis in our manuscript. In the revised of the manuscript, we have added some quantitative analysis and describe in the experimental verification and in both dam failures simulations according this comment on page 7, page 9 and page 16-21 (marked in blue).

**Comment 4:** Historical results are cited to confirm the goodness of the simulation, but no field data or observations are presented to validate the goodness of the modelling procedure.

**Response:** Indeed, there is lack of field data or observations in evaluating the simulation results of real case because the occurrence of tailings dam break is uncontrollable and some quantitative information in the process of dam break is difficult to collect. However, according to the satellite images after the disaster, we measured the impact range and image area of the Feijiao event in Brizial, which are very consistent with the simulation results,

**Comment 5:** It is not clear the cell size used in the terrain model and which DEM is used for the A'xi tailing dam event (pre or post event)?

**Response:** During the simulation for the A'xi tailings dam, we used two different DEMs (UAV DEM and ALOS DEM) with 0.5m × 0.5m and 12.5m × 12.5m resolutions, respectively. The UAV DEM is mosaiced into the ALOS DEM and forms a final DEM with of cell size of 0.5m × 0.5m resolution. But, the effective precision in the extent of ALOS DEM is still only 12.5m × 12.5m. We have supplied the relevant description on Page 14 of the revised paper.

**Comment 6:** The authors attributed some rheologic parameters to the modelled flows without performing a back analysis to define such parameters and to justify their choice.

**Response:** To justify our choice in rheological parameters, some discussions on the value of rheological parameters are supplemented on page14-15 of the revised version. In this study, the Bingham viscosity and yield stress were determined based on a previous investigation report conducted in testing the rheological properties of tailings using LAMY RM100 rotational viscometer (Liao and Zhou, 2015). Liao (2015) collected several tailings samples and conducted 15 sets of tailings fluid rheology tests, and the test operation complies with the rheological test specification. We selected the test results of one group of the iron ore tailings (Feijiao tailings dam is also an iron ore tailings pond) as rheological parameters for the test.

[revised manuscript text omitted]